# The prevalence, spatial distribution and geographic weighted regression of open defecation practice in sub-Saharan Africa using demographic and health survey (DHS) data

Lidetu Demoze[1]*, Natnael Gizachew[2], Eshetu Abera Worede[1], Adem Tsegaw Zegeye[3], Yayeh Walle Molalign[1], Mitkie Tigabie[4], Abiy Ayele Angelo[5], Eyob Akalewold Alemu[6], Gelila Yitageasu[1]

1 Department of Environmental and Occupational Health and Safety, Institute of Public Health, College of Medicine and Health Sciences, University of Gondar, Gondar, Ethiopia, 2 School of Public Health, College of Health Science and Medicine, Dilla University, Dilla, Ethiopia, 3 Department of Health Informatics, Institute of Public Health, College of Medicine and Health Sciences, University of Gondar, Gondar, Ethiopia, 4 Department of Medical Microbiology, School of Biomedical and Laboratory Science, College of Medicine and Health Sciences, University of Gondar, Gondar, Ethiopia, 5 Department of Immunology and Molecular Biology, School of Biomedical and Laboratory Sciences, College of Medicine and Health Sciences, University of Gondar, Gondar, Ethiopia, 6 Department of Epidemiology and Biostatistics, Institute of Public Health, College of Medicine and Health Sciences, University of Gondar, Gondar, Ethiopia

* lidetudemoze12@gmail.com

**Editor:** D. Daniel, Gadjah Mada University Faculty of Medicine, Public Health, and Nursing: Universitas Gadjah Mada Fakultas Kedokteran Kesehatan Masyarakat dan Keperawatan, INDONESIA

## Abstract

### Background

Open defecation is a harmful and unsafe practice that contributes to environmental pollution and disproportionately affects developing nations, particularly those in Sub-Saharan Africa. According to the World Health Organization (WHO) and the United Nations International Children's Emergency Fund (UNICEF) Joint Monitoring Programme (JMP), Sub-Saharan Africa is home to 46% of the global population still practising open defecation. Socio-economic factors, cultural norms, and individual attitudes play crucial roles in shaping sanitation behaviours and influencing open defecation practices. Therefore, this study aims to determine the prevalence, spatial distribution, and geographic inequalities of open defecation in Sub-Saharan Africa.

### Methods

A community-based cross-sectional survey was conducted, including 20,130 clusters and 496,957 households from 34 Sub-Saharan African countries. The Demographic and Health Survey (DHS) data were weighted, cleaned, and analyzed using Microsoft Excel, Stata version 17, ArcGIS version 10.7, and SaTScan™ version 10.1. Spatial analyses were performed using ArcGIS version 10.7 and Kulldorff's SaTScan™ version 10.1, while Geographically Weighted Regression (GWR) analyses were conducted using ArcGIS version 10.7.

**Data availability statement:** The data underlying this study are publicly available from the Demographic and Health Surveys (DHS) Program website at https://dhsprogram.com/data/. Access to the data requires free registration and approval from the DHS Program. The authors obtained the data after completing the registration process and adhering to the terms of data use. The authors do not have any special access privileges that others would not have. Interested researchers can access the same datasets in the same manner as the authors.

**Funding:** The author(s) received no specific funding for this work.

**Competing interests:** The authors have declared that no competing interests exist.

## Results

The prevalence of open defecation among households in Sub-Saharan Africa was 23.24% (95% CI: 23.12–23.35). The practice was clustered across enumeration areas (Global Moran's I = 0.25, Z-score = 366.12, P-value ≤ 0.001). The Getis-Ord Gi* statistic identified hotspots of open defecation primarily in East Africa, Central Africa, and West Africa. Anselin Local Moran's I detected both high and low clusters of open defecation, while SaTScan cluster analysis identified 146 windows containing significant clusters of households practising open defecation across Sub-Saharan Africa. Geographically Weighted Regression (GWR) analysis revealed that several factors were positively associated with open defecation, including lack of educational attainment, unimproved drinking water sources, lack of basic access to water, younger household heads, and extreme poverty. Additionally, household size greater than four, the richest households and urban and rural residency were negatively associated with open defecation practices.

## Conclusion

This study reveals a high prevalence of open defecation (23.24%) in Sub-Saharan Africa with significant geographic clustering, particularly in East, Central, and West Africa. This estimate is higher than the 18% reported by the 2021 WHO/UNICEF Joint Monitoring Programme (JMP). Novel spatial and GWR analyses uncovered associations with poverty, lack of education, water access, age of household heads, and wealth status. These findings underscore the need for geographically targeted, multi-sectoral sanitation interventions that address underlying socio-demographic disparities. Future research should explore the effectiveness of spatially tailored programs and integrate behavioral insights to accelerate progress toward Sustainable Development Goal 6.

## Introduction

A lack of access to safe sanitation is a global crisis [1]. The United Nations and United Nations Development Programme estimate that over 4.5 billion people lack access to safe sanitation [2]. In 2022, 420 million people worldwide practise open defecation, with sub-Saharan Africa having the highest prevalence among the seven regions, followed by South Asia [3].

While open defecation remains concentrated in sub-Saharan Africa and South Asia, the distribution has shifted significantly: in 2000, 67% of those practising it lived in South Asia (primarily India) and 17% in sub-Saharan Africa, whereas now those figures are 44% in South Asia and 47% in sub-Saharan Africa, respectively [3]. Although open defecation is practiced in nearly all regions of the world, it remains most widespread in India and several countries in sub-Saharan Africa, including Niger, Chad, Nigeria, Ethiopia, Kenya, Madagascar, Uganda, South Sudan, Burkina

Faso, and Togo [4]. Poor sanitation accounts for 10% of the global disease burden, contributing to diarrheal diseases, neglected tropical diseases, acute respiratory infections, and malnutrition, particularly among children [5–9].

Open defecation is a harmful and unsafe practice that causes pollution and disproportionately impacts developing nations, especially those in Asia and Sub-Saharan Africa [10]. 90% of people who engage in OD reside in rural areas of these regions [11,12]. The United Nations considers access to sanitation, along with clean drinking water, a basic human right, and Sustainable Development Goal 6.2 strives to eliminate open defecation by the year 2030 [13,14].

Several studies indicate that the prevalence of open defecation in Asian countries varies, with 27.7% in India, 16.8% in Nepal, 23.3% in urban communities in Indonesia, and 9% among women in Bangladesh [15–18]. Between 2000 and 2022, the global number of people practising open defecation declined from 1.3 billion to 420 million, a reduction of more than two-thirds [19]. However, the decline in open defecation has been less pronounced in sub-Saharan Africa, where the number of households practising it decreased only from 207 million in 2015–193 million in 2022, meaning that this region is home to 46% of the global population still practising it [20]. In addition, a study in 2022 reported that the pooled prevalence of open defecation practices among households in sub-Saharan African countries was 22.55% [21]. The population of sub-Sharan Africa exceeds 1.1 billion people, out of which, a large percentage is below the age of 20 [22]. Sub-Saharan Africa is economically rich due to its natural resources like minerals and oil, but many of the nations suffer from poverty and technological development [23]. Another major facet of the region's economy is its culture, which is composed of many ethnicities and languages stemming from Christianity and Islam [23].

Socio-economic factors, cultural norms, and individual attitudes including poverty, lack of access to sanitation facilities, place of residence, media exposure, access to drinking water, country income status, gender, and education all play crucial roles in shaping sanitation behaviours and influencing open defecation practices [20,21,24]. A previous study estimated the pooled prevalence and factors associated with open defecation practices in 33 sub-Saharan African countries using Demographic and Health Survey data up to 2020 [21]. However, the study did not examine the spatial variation of open defecation practices or how the relationships between predictors and open defecation vary across locations.

Additionally, several DHS datasets have been released after 2020, providing a valuable opportunity to assess the most recent status of open defecation practices in sub-Saharan Africa. Moreover, spatial analysis helps to identify hotspot areas of open defecation and provides insights that enable public health programmers and policymakers to engage in strategic planning. Therefore, this study aims to analyze the spatial distribution and associated factors with open defecation practices among households in Sub-Saharan Africa using DHS data.

## Materials and methods

### Study design and settings

A nationwide community-based cross-sectional study design was conducted in each country. A total of 34 sub-Saharan African countries (Angola, Benin, Burkina Faso, Burundi, Cameroon, Chad, Comoros, the Democratic Republic of Congo, Côte d'Ivoire, Ethiopia, Gabon, Gambia, Ghana, Guinea, Kenya, Lesotho, Liberia, Madagascar, Malawi, Mali, Mauritania, Mozambique, Namibia, Niger, Nigeria, Rwanda, Senegal, Sierra Leone, South Africa, Tanzania, Togo, Uganda, Zambia, and Zimbabwe) were included in this study (**Fig 1**). The term, sub-Saharan Africa, refers to the expanse of the African continent that is located south of the Sahara Desert [25]. This region includes savanna, rainforests, and mountain ranges. Approximately forty-nine countries, including Nigeria, Ethiopia, Kenya, and South Africa, make up this region that stretches over 9.3 million square miles (24 million square kilometres) in area [26,27].

### Data source and study period

This study utilized publicly available secondary data from the Demographic and Health Surveys (DHS) Program, accessible at https://dhsprogram.com. The DHS Program provides nationally representative household survey data on various health and demographic indicators. The data used in this study ranges from 2012 to 2023/24 (**Table 1**).

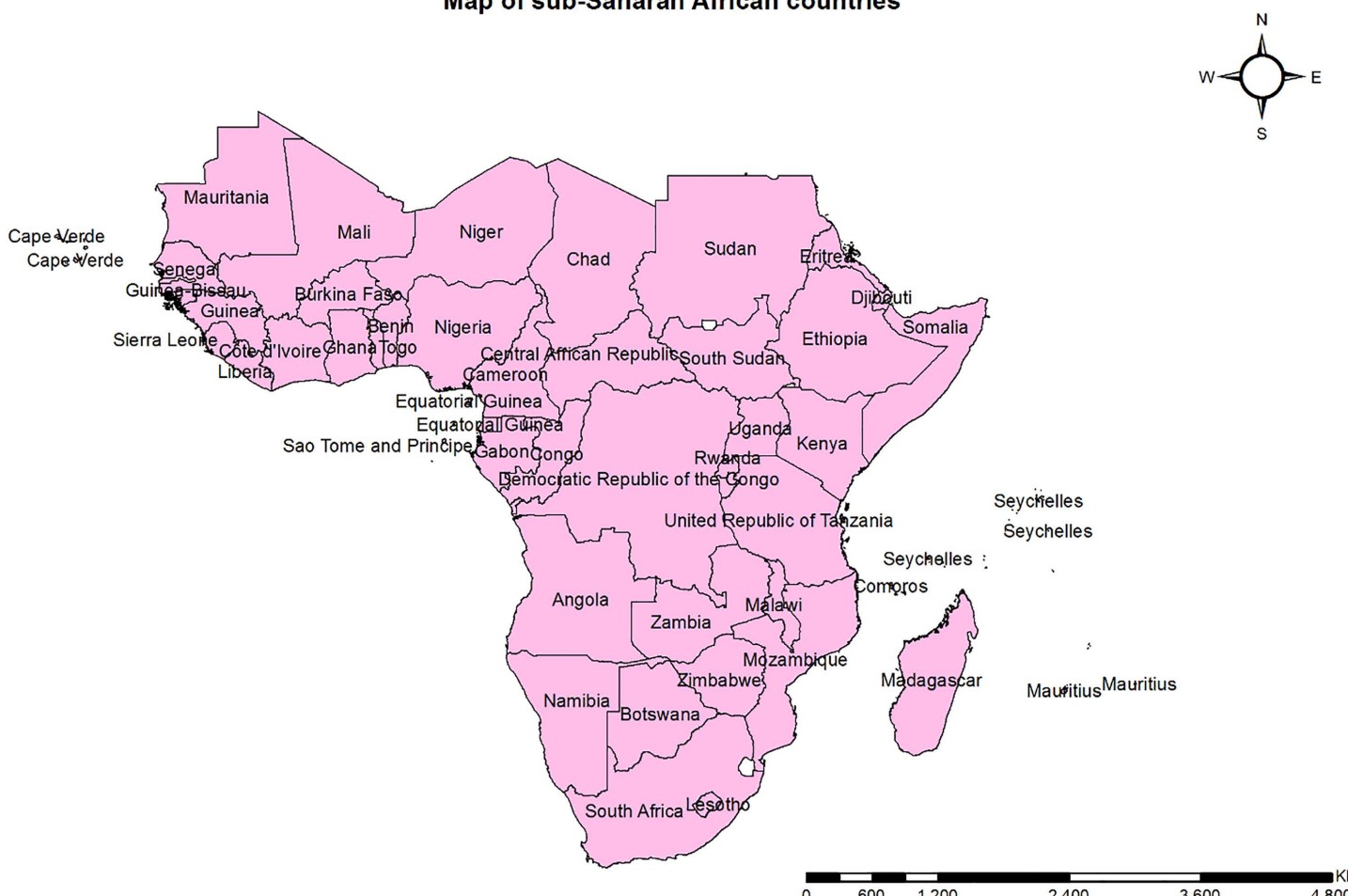

**Source of shape file: Global Administrative Areas Database (GADM), own map output from ArcGIS V.10.7**

**Fig 1. The study area map.**

## Population and eligibility criteria

The source population for this study comprised all households residing in urban and rural areas of 34 sub-Saharan African countries at the time of the survey. The study population included all households from the selected enumeration areas (clusters) in both urban and rural settings that were included in the analysis. We included only clusters and households that had complete data on the outcome variable and selected explanatory variables. Clusters with missing spatial coordinates or implausible geographic values were excluded. The analysis was based on Household Recode (HR) data from DHS to ensure uniformity of household-level variables.

## Sample size and sampling techniques

A total of 20,130 clusters and 496,957 households from 34 Sub-Saharan African countries were included in the analysis for this study. The Demographic and Health Surveys (DHS) Program applies a rigorous two-stage stratified cluster sampling design to obtain national representativeness with data collection efficiency. The population is initially stratified based

**Table 1. List of countries and corresponding DHS survey years included in the analysis.**

| Country | DHS year |
|---|---|
| Angola | 2015−16 |
| Benin | 2017−18 |
| Burkina Faso | 2021 |
| Burundi | 2016−17 |
| Cameroon | 2018 |
| Chad | 2014–2015 |
| Comoros | 2012 |
| Congo Democratic Republic | 2013−14 |
| Cote d'Ivoire | 2021 |
| Ethiopia | 2016 |
| Gabon | 2019−21 |
| Gambia | 2019−20 |
| Ghana | 2022 |
| Guinea | 2018 |
| Kenya | 2022 |
| Lesotho | 2023−24 |
| Liberia | 2019−20 |
| Madagascar | 2021 |
| Malawi | 2015−16 |
| Mali | 2018 |
| Mauritania | 2019−21 |
| Mozambique | 2022−23 |
| Namibia | 2013 |
| Niger | 2012 |
| Nigeria | 2018 |
| Rwanda | 2019−20 |
| Senegal | 2023 |
| Sierra Leone | 2019 |
| South Africa | 2016 |
| Tanzania | 2022 |
| Togo | 2022 |
| Uganda | 2016 |
| Zambia | 2018 |
| Zimbabwe | 2015 |

on geographic or administrative divisions and rural-urban categories before sampling for balanced representation. The enumeration of the Enumeration Areas (EAs) or clusters probability proportionate to size (PPS) is conducted at the initial stage by using the recent national census.

This provides both the rural and urban population an adequate representation. Household listing activity within every EA is conducted in the second stage to update the list of all the households. A specified number of households, typically 25–30 per EA, is randomly selected through systematic sampling. All the eligible individuals from the sampled households are invited for the survey [28]. Furthermore, the DHS Program applies a consistent methodology in countries, allowing for comparison of data collection and analysis. By this systematic approach, DHS surveys provide high-quality, nationally representative data for key health and demographic indicators.

## Study variables

**Dependent variable.** The outcome variable in this study was open defecation practice, as defined by the WHO/UNICEF Joint Monitoring Programme (JMP) [29,30]. It was measured as "Yes" for households practising open defecation (i.e., having no sanitation facility or defecating in a bush or field) and "No" for those with access to basic sanitation facilities.

**Independent variables.** Independent variables include sociodemographic, economic and water-related characteristics of households (Table 2).

## Data management and analysis

Data extraction, coding, and analysis were done using Microsoft Excel, Stata version 17, Arc-GIS version 10.7 and SaTS-can™ version 10.1 statistical software. Missing data were handled using complete-case analysis, where only households with complete information on the outcome variable and selected explanatory variables were included in the final analysis. This approach aligns with DHS analytic guidelines and helps ensure consistency and comparability across countries [31]. The dependent variable, open defecation, was coded as 'Yes' if the household reported having no facility/bush/field and 'No' for any other facility type. Independent variables such as age of household head, educational attainment, household size, media exposure, lace of residence, access to water source, water source, and wealth index were categorized based on standard DHS definitions and supported by previous literature [21,29]. Variables were recoded to ensure consistency

**Table 2. A list of independent variables or predictors of open defecation practice.**

| Variables | Category |
|---|---|
| Age of household head | 11-35 |
| | 36-50 |
| | >50 |
| Sex of household head | Male |
| | Female |
| Household educational attainment | No education |
| | Primary |
| | Secondary |
| | Higher and above |
| Wealth index | Poorest |
| | Poorer |
| | Middle |
| | Richer |
| | Richest |
| Media exposure | No |
| | Yes |
| Family size | ≤ 4 |
| | > 4 |
| Place of residence | Rural |
| | Urban |
| Source of drinking water source | Improved |
| | Unimproved |
| Access to water source | Basic (≤ 30) |
| | Not basic (> 30) |

 

across countries. Before analysis, sampling weights were calculated for each variable to adjust for oversampling or under-sampling and to ensure that the estimates remained nationally representative.

**Spatial analysis.** Global spatial autocorrelation was conducted to determine whether the spatial distribution of open defecation among households in Sub-Saharan Africa was dispersed, clustered, or randomly distributed. Global Moran's I is a spatial statistic used to measure spatial autocorrelation by analyzing the entire dataset and producing a single output value ranging from −1 to +1. A Moran's I value close to −1 indicates that open defecation practices among households are dispersed, while a value near +1 signifies clustering. A Moran's I value close to 0 suggests that open defecation practices are randomly distributed [30,32,33].

To explore variations in spatial autocorrelation within the study area, a hotspot analysis was conducted using the Getis-Ord Gi* statistic. This involved calculating the Gi* statistic for each geographic unit. The statistical significance of observed clustering was evaluated using z-scores and p-values. Specifically, areas with high Gi* values were identified as hotspots, representing statistically significant clusters of high values, whereas areas with low Gi* values were designated as coldspots, indicating statistically significant clusters of low values [32,34].

Cluster and Outlier Analysis using Anselin Local Moran's I in ArcGIS was conducted. Household-level data were spatially referenced and analyzed, classifying locations into high-high (HH) clusters, low-low (LL) clusters, high-low (HL) outliers, and low-high (LH) outliers. HH and LL clusters indicated consistent high or low rates, while HL and LH outliers revealed deviations from surrounding patterns [29,35,36].

SaTScan cluster analysis was performed to identify high-likelihood clusters of open defecation practices using the Bernoulli model, where households practising open defecation were classified as cases and non-practising households as controls [37]. The scan statistic detected clusters with significantly high prevalence through a circular scanning window that assessed varying cluster sizes. Likelihood ratio tests and Monte Carlo simulations were used to determine statistical significance. The results identified high-risk areas, providing insights for targeted sanitation interventions and resource allocation [37,38].

**Spatial regression analysis.** To examine the relationship between open defecation practices and potential predictor variables, a spatial regression analysis was conducted using the Ordinary Least Squares (OLS) method and Geographically Weighted Regression. OLS regression was applied to assess how various socioeconomic, environmental, and demographic factors influence open defecation prevalence.

$Y = \beta_0 + \beta_1 X_1 + \beta_2 X_{2} + {}_{...} \beta_n X_n + \varepsilon$; where: Y = dependent variable(open defecation); $\beta_0$ = intercept; $\beta_1, \beta_2... \beta_n$ = coefficients; X1, X2…$X_n$ = explanatory variables; $\varepsilon$ = residual [29].

The OLS depends on these assumptions. First, each explanatory variable was expected to exhibit the anticipated relationship and be statistically significant, either positively or negatively. Residuals should not be spatially clustered, which was tested using the global spatial autocorrelation tool (Global Moran's I). The Jarque-Bera test was used to assess whether the residuals followed a normal distribution. Before running the GWR model, multicollinearity among the explanatory variables was assessed using the Variance Inflation Factor (VIF) in the OLS model, with all values falling below the threshold of 7.5, indicating no serious multicollinearity. Since GWR inherits variables from the OLS model, and no new predictors are introduced, this assessment was considered sufficient for the GWR model as well.. Finally, the performance of the OLS model was evaluated using the adjusted $R^2$ [29,39].

Geographically Weighted Regression (GWR) is a spatial regression technique that extends Ordinary Least Squares (OLS) by allowing relationships between predictors and the dependent variable to vary across geographic space. Unlike traditional regression models, which assume a global relationship, GWR captures local variations in predictors' influence on open defecation prevalence [7]. Adjusted $R^2$ and Akaike Information Criterion (AIC) were used to compare GWR with OLS. Local coefficient maps visualized spatial variations in predictor effects [7].

Clusters and households were included in the analysis if they had complete data on the outcome variable (open defecation status) and all selected explanatory variables. The analysis was conducted using the Household Recode (HR) files

from the DHS datasets. Clusters with missing spatial coordinates or implausible values were excluded to maintain spatial accuracy. Prior to fitting the Geographically Weighted Regression (GWR) model, we first conducted an exploratory regression to screen potential predictors of open defecation. This was followed by Ordinary Least Squares (OLS) regression to assess the global relationships, significance, and multicollinearity among variables. Only variables that met statistical criteria (p < 0.05) and had acceptable multicollinearity levels (VIF < 7.5) were included in the final GWR model, as recommended in spatial modeling guidelines [40–43]. For the GWR model, an adaptive bandwidth was selected to account for spatial variation in observation density across the study area. The bandwidth choice was determined automatically using the Akaike Information Criterion corrected (AICc), which optimizes model fit by balancing bias and variance [44]. Adaptive bandwidths are appropriate when the spatial distribution of observations is uneven, as is typical with Demographic and Health Survey (DHS) data collected across diverse geographic settings [45].

### Ethical approval and consent to participate

This study is a secondary analysis of previously collected and published household survey data. As such, it did not require independent ethical approval. However, the DHS program obtained ethical approval for each country's survey from relevant national Institutional Review Boards (IRBs). All DHS surveys included in this analysis received ethical clearance and informed consent was obtained from all participants at the time of original data collection. Therefore, ethical approval and informed consent do not apply to this study.

## Results

### Socio-demographic and water-related characteristics of participants

This study used a weighted sample of 496,957 households across 20,130 clusters (S1 File). The age of the household head was categorized into three groups: 11–35 years (33.49%), 36–50 years (32.94%), and above 50 years (33.57%). The sex of the household head indicates that the majority were male (71.70%), while females accounted for 28.30%. Regarding household educational attainment, 32.77% of household heads had no formal education, 31.22% had attained primary education, 26.86% had secondary education, and only 9.15% had higher education.

The wealth index classifies households into five economic categories: poorest (19.40%), poorer (19.37%), middle (19.62%), richer (20.68%), and richest (20.93%). Media exposure indicates that 22.10% of households lack access to media, while 77.90% have exposure. In terms of family size, 51.51% of households have four or fewer members, while 48.49% have more than four. The place of residence shows that 61.09% of households are in rural areas, whereas 38.91% are urban dwellers. For drinking water sources, 39.11% of households rely on improved water sources, while 60.89% use unimproved sources. Lastly, regarding access to water, 82.59% of households have basic access (≤30 minutes), while 17.41% have non-basic access (>30 minutes) (Table 3).

### The prevalence of open defecation practice in sub-Sharan Africa

The prevalence of open defecation practice in sub-Saharan Africa was 23.24% (95% CI: 23.12–23.35) (Fig 2).

### Spatial distribution of open defecation practices in sub-Saharan Africa

The map depicts the distribution of open defecation practices in Sub-Saharan Africa. Areas coloured red indicate a higher prevalence of open defecation, while areas coloured black represent the lowest prevalence (Fig 3).

The Global Moran's I statistic used to assess the spatial autocorrelation of open defecation practice in Sub-Saharan Africa indicates that the distribution of open defecation is significantly clustered rather than randomly distributed. The Moran's Index (I) of 0.25 suggests spatial clustering, meaning that areas with high or low open defecation rates are located near similar areas. The expected index (−0.000050) represents the value anticipated under complete spatial

**Table 3. Socio-demographic and water-related characteristics of households in sub-Saharan Africa.**

| Variables | Category | Frequency(n) | Percent (%) |
|---|---|---|---|
| Age of household head | 11-35 | 166427 | 33.49 |
| | 36-50 | 163680 | 32.94 |
| | >50 | 166850 | 33.57 |
| Sex of household head | Male | 356323 | 71.70 |
| | Female | 140634 | 28.30 |
| Household educational attainment | No education | 162871 | 32.77 |
| | Primary | 155163 | 31.22 |
| | Secondary | 133460 | 26.86 |
| | Higher and above | 45463 | 9.15 |
| Wealth index | Poorest | 96392 | 19.40 |
| | Poorer | 96238 | 19.37 |
| | Middle | 97519 | 19.62 |
| | Richer | 102758 | 20.68 |
| | Richest | 104050 | 20.93 |
| Media exposure | No | 109847 | 22.10 |
| | Yes | 387110 | 77.90 |
| Family size | ≤ 4 | 255975 | 51.51 |
| | > 4 | 240982 | 48.49 |
| Place of residence | Rural | 303615 | 61.09 |
| | Urban | 193342 | 38.91 |
| Source of drinking water source | Improved | 194360 | 39.11 |
| | Unimproved | 302597 | 60.89 |
| Access to water source | Basic (≤ 30) | 410414 | 82.59 |
| | Not basic (> 30) | 86543 | 17.41 |

randomness, and since the observed value is much higher, it confirms a non-random pattern. The variance (0.000000) is extremely small, indicating high precision in the autocorrelation results.

Additionally, the z-score (366.12) is exceptionally high, far exceeding the critical threshold of ±2.58, signifying strong clustering with high statistical significance. The p-value (0.000000), being less than 0.05, further confirms that this clustering pattern is not due to random chance. The visualization of the z-score distribution reinforces these findings, as the value falls within the significant clustered region (>2.58). Areas with a high prevalence of open defecation tend to be located near other high-prevalence areas, while regions with low prevalence are also clustered together.

This analysis was conducted using specific spatial parameters, including the Inverse Distance conceptualization (where closer features exert a stronger influence), the Euclidean distance method (which calculates straight-line distances), and a distance threshold of 1.2858 meters, defining the neighbourhood relationships (**Fig 4**).

Hot spot areas of open defecation were identified in East Africa, Central Africa, and West African countries, marked by dark red and light red shades. In contrast, cold spots were observed across all Sub-Saharan African countries, represented by dark blue and light blue shades. The Cluster and Outlier analysis (Anselin Local Moran's I) reveals spatial patterns in open defecation practices in sub-Saharan Africa. High-high (HH) clusters (yellow) indicate areas with persistently high prevalence of open defecation practice, requiring urgent intervention. Low-low (LL) clusters (black) reflect effective sanitation systems with consistently low rates. High-low (HL) outliers (red) highlight the unexpectedly high prevalence of open defecation practice in otherwise low-rate areas, often linked to localized issues. Low-high (LH) outliers (blue) represent well-managed sanitation zones within high-prevalence regions, offering best-practice models.

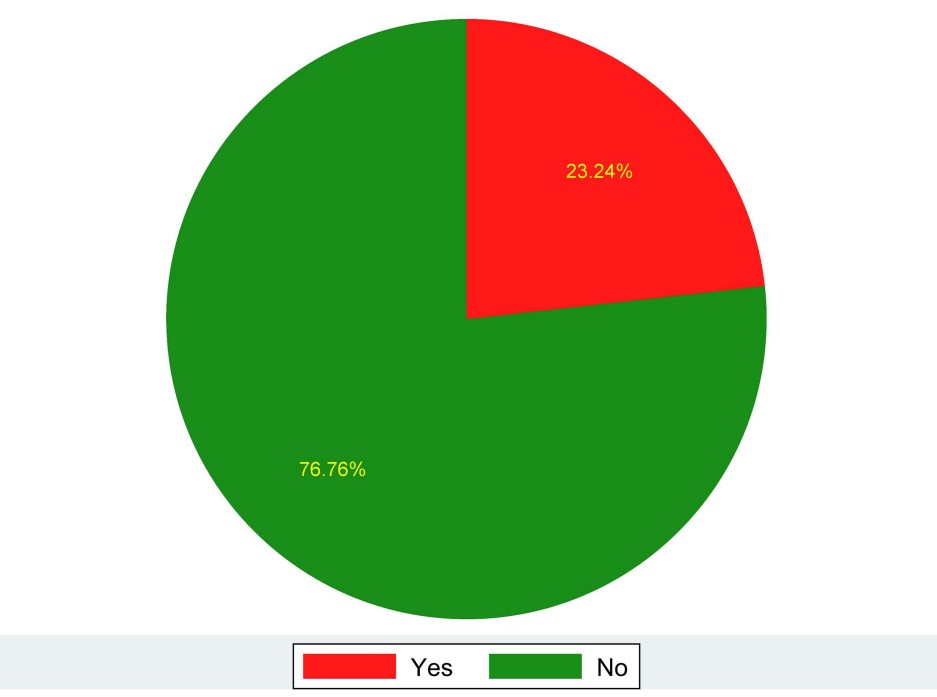

**Fig 2. The prevalence of open defecation practice in sub-Saharan Africa.**

The SaTScan cluster analysis identified 146 windows containing clusters among 496,957 households in Sub-Saharan Africa. Significant clusters were observed across nearly all regions of Sub-Saharan Africa. The shaded areas on the map represent varying levels of statistical significance based on Log Likelihood Ratio (LLR) values, indicating the clustering of open defecation practices. Regions with LLR values between 7.42 and 307.55 exhibit the least statistically significant clusters, corresponding to lower concentrations of open defecation. Areas with moderate significance, ranging from 307.56 to 1020.22, show an increasing prevalence of open defecation. Clusters with higher significance (LLR values between 1020.23 and 2814.04) indicate a more pronounced pattern of open defecation clustering. Very high significance is observed in areas with LLR values between 2814.05 and 5659.38, where open defecation is more concentrated. Finally, the most statistically significant clusters, with LLR values between 5659.39 and 9025.88, highlight regions with the highest prevalence of open defecation (**Fig 5a–c**).

## Spatial regression analysis

The Ordinary Least Squares (OLS) regression analysis was conducted to examine the determinants of open defecation in households. The model explains 40% of the variance in open defecation practices (Adjusted $R^2 = 0.40$). The joint F-statistic (1,349.68, $p < 0.001$) indicates that the overall model is statistically significant. Similarly, the Wald chi-squared test (2,469.27, $p < 0.001$) further confirms the robustness of the model. The Variance Inflation Factor (VIF) values ranged from 1.50 to 5.46, indicating moderate correlation (below 7.5) among some independent variables. However, the level of multi-collinearity is not severe enough to justify excluding any variables. The Koenker BP test ($\chi^2 = 96.23$, $p < 0.001$) confirms the presence of heteroskedasticity. The Jarque–Bera test ($\chi^2 = 12.94$, $p < 0.001$) indicates that the residuals deviate from normality. However, this deviation is common in large sample sizes and does not necessarily invalidate the model's findings. Therefore, we opted to fit a Geographically Weighted Regression (GWR) analysis because of the spatial non-stationary or heteroskedasticity in the Koenker (BP) statistic test. Since the Koenker (BP) statistic indicated statistical significance,

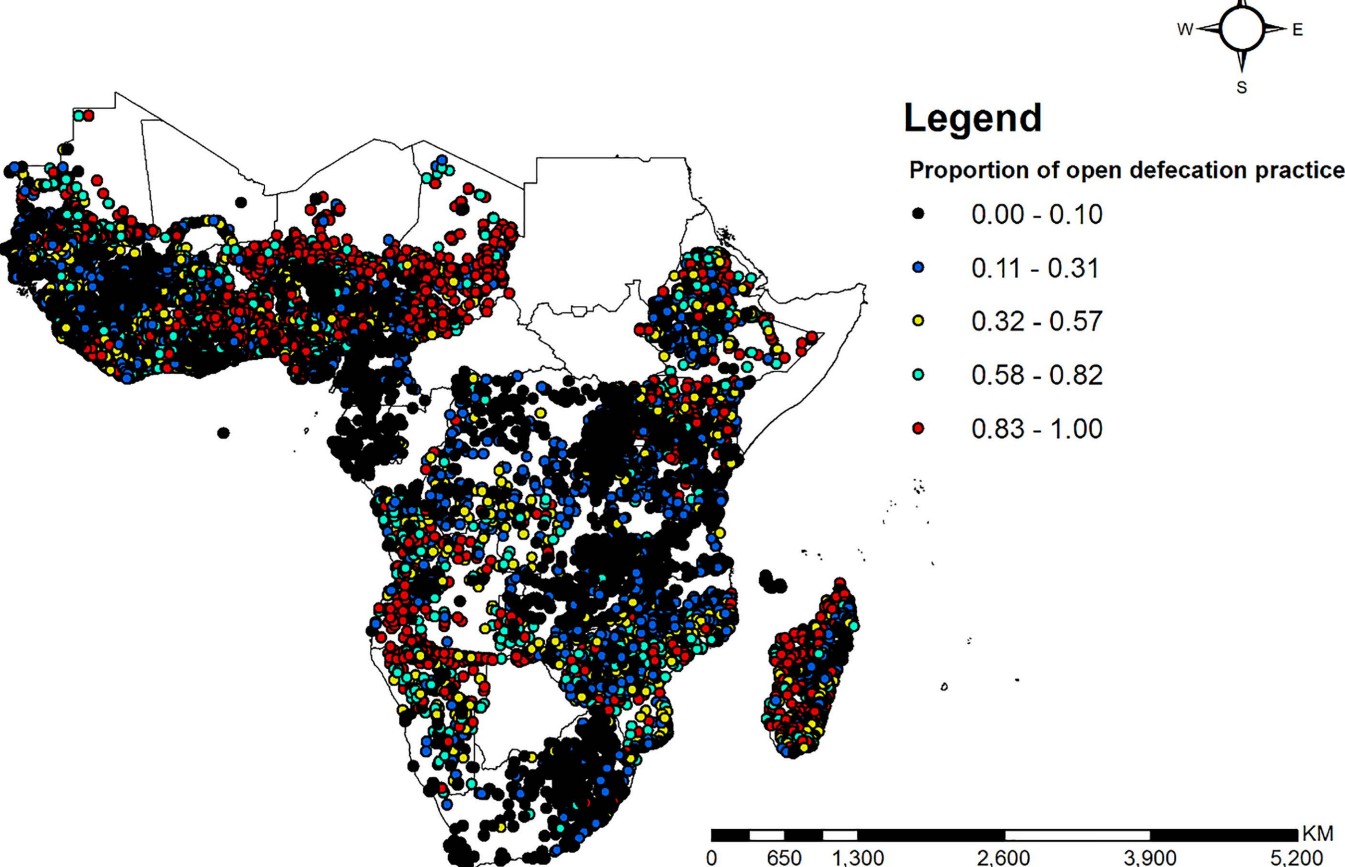

**Spatial variation of open defecation practice in sub-Saharan Africa**

Legend

**Proportion of open defecation practice**

- • 0.00 - 0.10
- • 0.11 - 0.31
- • 0.32 - 0.57
- • 0.58 - 0.82
- • 0.83 - 1.00

**Source of shape file:Global Administrative Areas Database (GADM),own map output from ArcGIS V.10.7**

**Fig 3. The spatial distribution of open defecation practice in sub-Saharan Africa.**

we relied on robust probability values to assess the significance of the coefficients. Households with no formal education ($\beta = 0.34$, $p < 0.001$), larger family sizes (greater than four members) ($\beta = -0.04$, $p = 0.02$), urban households ($\beta = -0.04$, $p = 0.02$), and rural households ($\beta = -0.03$, $p = 0.04$) were all significantly associated with open defecation in sub-Saharan Africa. Additionally, households with limited water access ($\beta = 0.11$, $p < 0.001$), those headed by individuals aged between 11–35 ($\beta = 0.08$, $p < 0.001$), the poorest households ($\beta = 0.32$, $p < 0.001$), and the richest households ($\beta = -0.07$, $p < 0.001$) also showed significant associations with open defecation practice. The local spatial regression (Geographically weighted regression) analysis showed that there was a significant improvement over the global ordinary least square regression analysis. The adjusted R-squared value increased from 40% in the OLS analysis to 72.42% in the GWR analysis, demonstrating that the GWR model more effectively captures the geographical variation of open defecation among households in Sub-Saharan Africa. Additionally, the higher AIC value in the OLS model (AIC = 139,182.78) compared to the GWR model (AIC = 123,788.24) further confirms that the GWR model provides a superior fit for the data (**Table 4**).

The results of the geographically weighted regression indicate a positive relationship between households with no formal education and open defecation practices. As the number of household heads without formal education increases, the prevalence of open defecation also rises. Areas with the highest positive correlation between open defecation practice

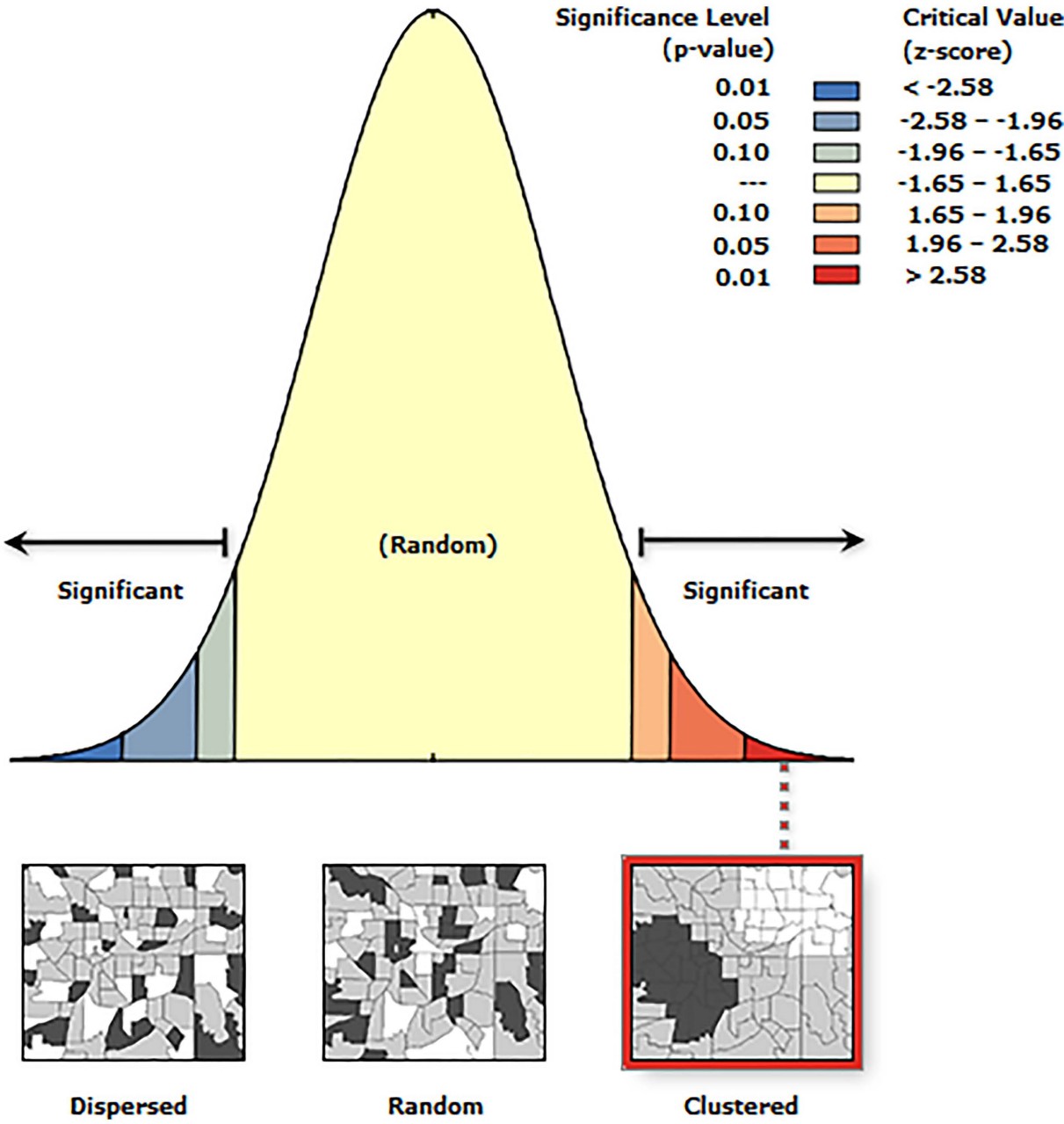

**Fig 4. The spatial autocorrelation result of open defecation practice in sub-Saharan Africa.**

and lack of education were found in Chad, Nigeria, Ghana, Niger, the Democratic Republic of the Congo, Ethiopia, Kenya, Namibia, and Guinea. The likelihood of open defecation practice decreases as the number of households with a family size greater than four rises. These areas include South Africa, Mozambique, the Democratic Republic of the Congo, Chad, Cameroon, Nigeria, Burkina Faso, Ghana, Niger, Togo, Benin, and Ethiopia. The likelihood of open defecation practices also increases as the number of households with unimproved drinking water rises. Areas with the highest positive correlation between open defecation practice and households with unimproved drinking water were observed in Comoros and Madagascar (**Fig 6a–c**).

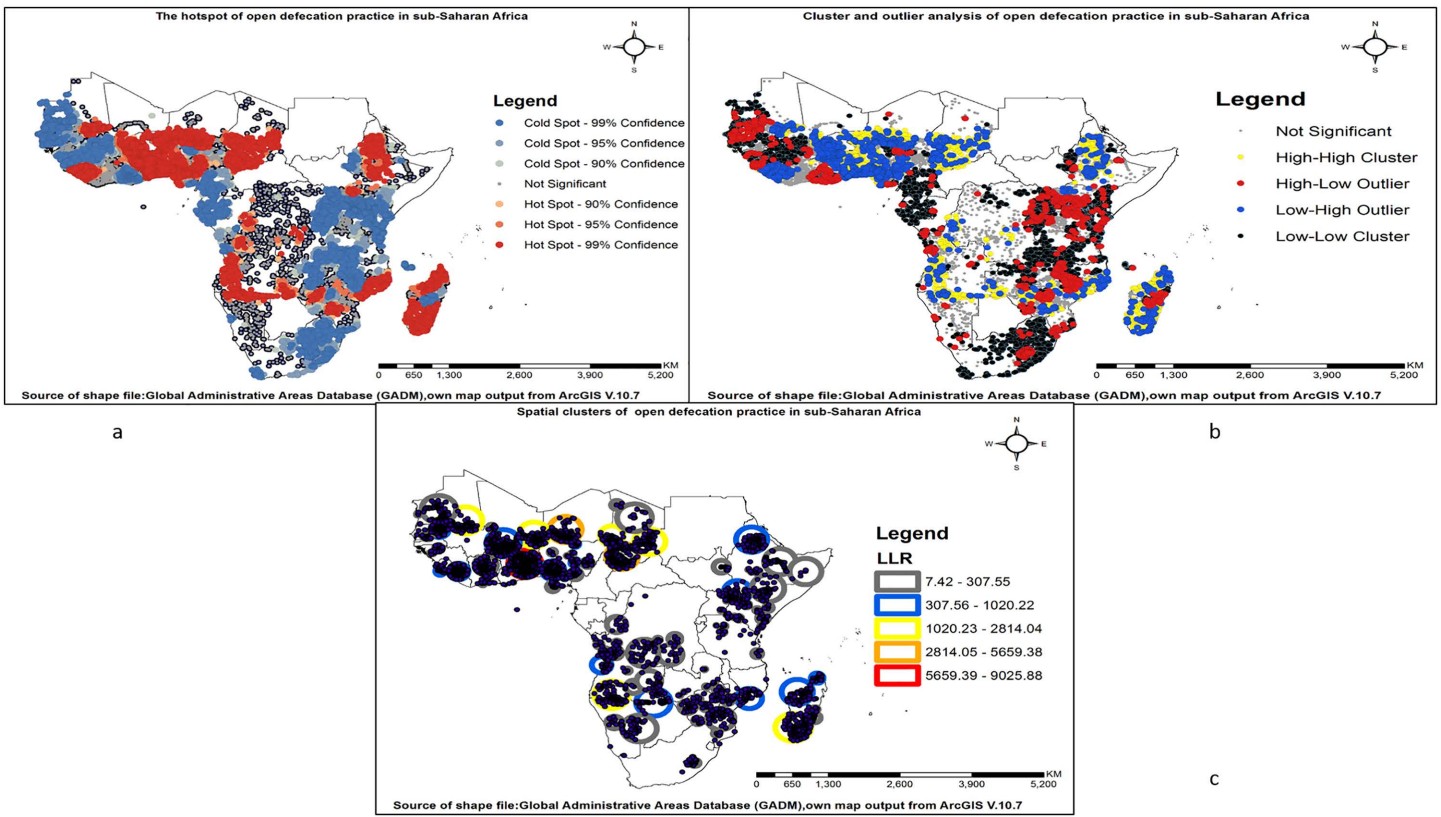

**Fig 5. Spatial clustering of open defecation practice among households in sub-Saharan Africa: (a) Hotspot and coldspot analysis; (b) Cluster and outlier analysis; and (c) SaTScan cluster analysis.**

The likelihood of open defecation practices increases as the number of households without basic access to a water source rises. Areas with the highest positive correlation between open defecation practice and households lacking basic access to a water source were observed in South Africa, Namibia, Nigeria, Cameroon, Ghana, Benin, Togo, Mali, Mauritania, Senegal, and Gambia. The likelihood of open defecation practice increases as the number of households headed by individuals aged 11–35 rises. Areas with the highest positive correlation between open defecation practice and households headed by individuals in this age group were observed in Angola, the Democratic Republic of the Congo, Chad and Cameroon. The likelihood of open defecation practice increases as the number of poorest households rises. Areas with the highest positive correlation between open defecation practice and the poorest households were observed in Ethiopia, Madagascar, Mozambique, Zimbabwe, Zambia, Angola, Namibia, Nigeria, Mauritania, Mali, Côte d'Ivoire, and Sierra Leone (**Fig 7a–c**).

The likelihood of open defecation practice decreases as the number of richest households rises. Areas with the highest negative correlation between open defecation practice and the richest households were observed in Nigeria, Togo, Benin, Burkina Faso, and Ghana. The likelihood of open defecation decreases as the number of households in urban areas increases. The strongest negative correlations between open defecation and urban households were observed in Ethiopia, Madagascar, Chad, Tanzania, the Democratic Republic of the Congo, Cameroon, and Niger. The likelihood of open defecation decreases as the number of rural households increases. The strongest negative correlations between open defecation and rural households were observed in Ethiopia, Madagascar, the Democratic Republic of the Congo, and Cameroon (**Fig 8a–c**).

**Table 4. Summary of OLS and GWR model results and diagnostics for open defecation practice among households in sub-Saharan Africa.**

| Variable | Coefficient | Standard error | t-statistics | Probability | Robust SE | Robust Probability P-value | VIF |
|---|---|---|---|---|---|---|---|
| Intercept | 0.12 | 0.088702 | 1.38 | 0.16 | 0.19 | 0.03* | ---- |
| No education | 0.34 | 0.007883 | 43.89 | 0.00* | 0.06 | < 0.001* | 1.97 |
| Higher education | 0.01 | 0.015677 | 0.15 | 0.88 | 0.10 | 0.90 | 2.57 |
| Family size >4 | −0.04 | 0.010647 | −4.47 | 0.00* | 0.02 | 0.02* | 4.40 |
| Unimproved drinking water source | 0.11 | 0.005398 | 20.87 | 0.00* | 0.01 | < 0.001* | 2.72 |
| No basic access to water | 0.05 | 0.008997 | 5.87 | 0.00* | 0.02 | < 0.01* | 1.50 |
| HH age between 11–35 | 0.08 | 0.012103 | 6.91 | 0.00* | 0.02 | < 0.001* | 4.00 |
| Poorest HH | 0.32 | 0.009544 | 34.06 | 0.00* | 0.02 | < 0.001* | 1.77 |
| Richest HH | −0.07 | 0.008827 | −8.80 | 0.00* | 0.01 | < 0.001* | 3.36 |
| Urban | −0.04 | 0.008769 | −4.92 | 0.00* | 0.02 | 0.02* | 5.46 |
| Rural | −0.03 | 0.008899 | −3.41 | 0.00* | 0.01 | 0.04* | 3.94 |

**Ordinary Least Square regression model diagnostic test**

| | | | |
|---|---|---|---|
| Number of observations | 20,130 | Adjusted R-squared | 0.40 |
| Joint F-statistics | 1349.68 | Prob(>F), (10, 20119) degree of freedom | < 0.001* |
| Joint Wald statistics | 2469.27 | Prob (> chi-squared), (10) degree of freedom | < 0.001* |
| Koenker (BP) statistics | 96.23 | Prob (> chi-squared), (10) degree of freedom | < 0.001* |
| Jarque–Bera statistics | 12.94 | Prob (> chi-squared), (2) degree of freedom | < 0.001* |

* = Significant independent variable at p-value less than 0.05
SE = Standard error
VIF = Variance Inflation Factor

**Summary of geographically weighted regression (GWR) analysis**

| Parameter | Value |
|---|---|
| AIC | 123788.23 |
| R² | 72.90 |
| Adjusted R² | 72.42 |
| Sigma | 5.20 |

## Discussion

This study identified open defecation as a major public health issue in sub-Saharan Africa, with an estimated prevalence of 23.24% (95% CI: 23.12–23.35). This estimate suggests that open defecation is a widespread practice in sub-Saharan Africa. The high prevalence underscores the urgent need for improved sanitation infrastructure and public health interventions to reduce open defecation and its associated health risks, such as diarrheal diseases and malnutrition.

This finding falls within the range reported by a previous study conducted in 33 sub-Saharan African countries, which estimated a pooled prevalence of 22.55% (95% CI: 17.49–27.61%) among households in the region [21]. The similar prevalence rates of open defecation in both studies can be attributed to their use of standardized DHS data, ensuring consistency and comparability across countries. By analyzing multiple sub-Saharan African nations, both studies capture regional trends rather than localized variations. Additionally, common structural factors such as poverty, inadequate sanitation, limited water access, and urban-rural disparities contribute to comparable prevalence estimates. Additionally, the prevalence is lower than studies conducted in Haiti (25.3%) and Indonesia (32.5%) [46,47]. The lower prevalence of open defecation in sub-Saharan Africa compared to Haiti and Indonesia is influenced by economic development, cultural

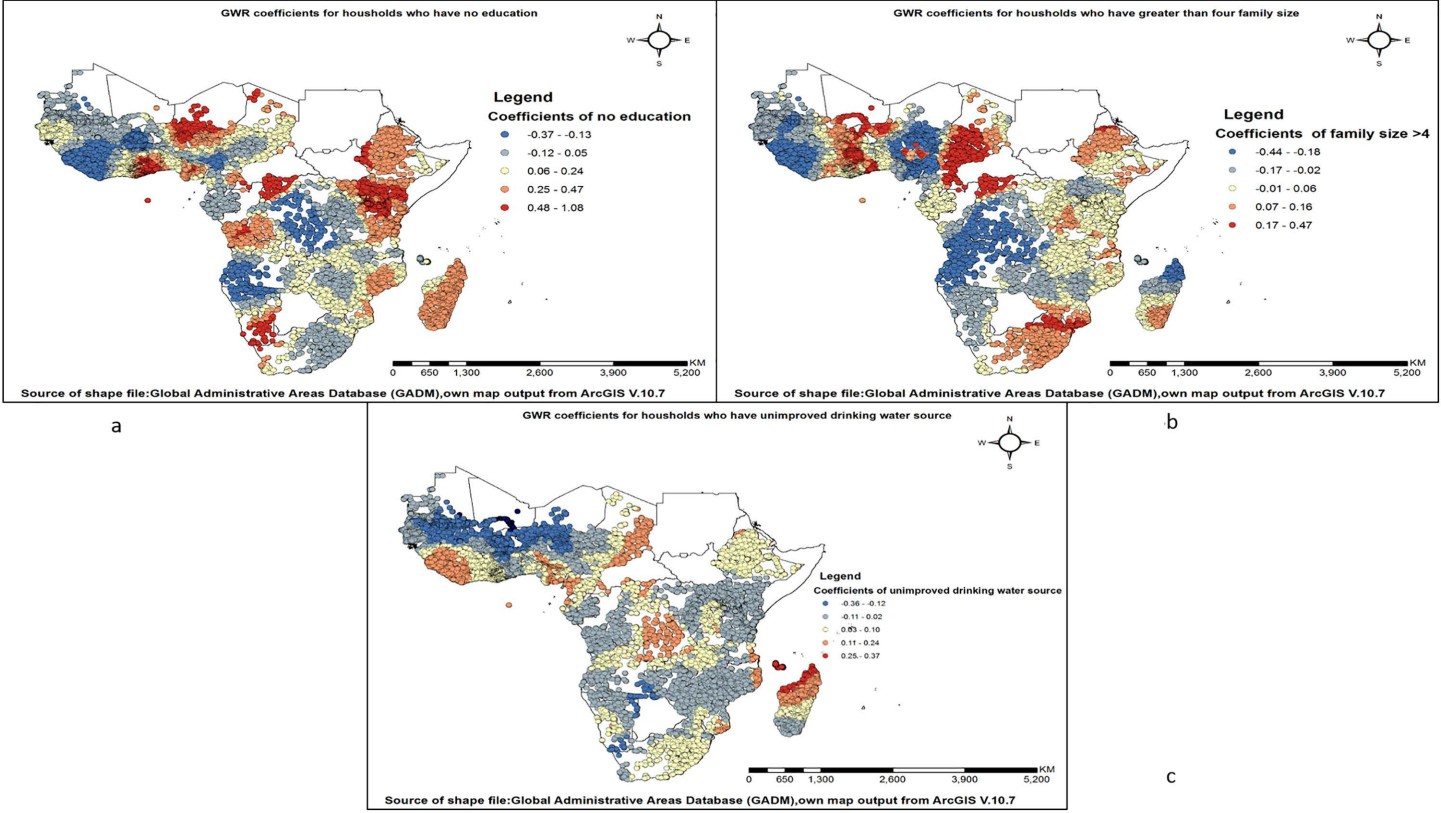

**Fig 6. Spatial variation in predictors associated with open defecation practice among households in sub-Saharan Africa.** (a) Coefficients for households with no education; (b) Coefficients for households with family size greater than four; and (c) Coefficients for households using unimproved drinking water sources.

norms, government policies, and urbanization. Better sanitation infrastructure, effective programs, and higher urbanization levels contribute to reduced open defecation rates [48,49]. In addition, many countries in sub-Saharan Africa have seen increased implementation of community-based sanitation programs and targeted hygiene campaigns in recent years, which may have contributed to relatively lower OD rates despite persistent challenges [50,51].

The estimate is higher than the 2021 Joint Monitoring Program (JMP) report by WHO and UNICEF, which reported an open defecation prevalence of 18% in sub-Saharan Africa, and the World Bank report, which estimated it at 11.1% in India [52,53]. The discrepancy in open defecation prevalence estimates between our study and reports like the 2021 Joint Monitoring Programme (JMP) by WHO/UNICEF or the World Bank may be due to several factors. Differences in geographic focus, with our study potentially covering high-prevalence areas, can influence estimates. Additionally, timing differences in data collection, cultural practices, sanitation infrastructure gaps, and the inclusion of underrepresented populations, such as those in informal settlements or conflict-affected regions, could further explain the higher prevalence reported in our study [24,54–55]. In addition, India has implemented extensive sanitation initiatives, such as the Swachh Bharat Abhiyan (Clean India Mission), which have significantly reduced open defecation rates [56].

This study revealed spatial variations in open defecation across clusters in Sub-Saharan Africa, as indicated by the spatial autocorrelation results. The Getis-Ord Gi* statistic identified hotspots of open defecation practice mainly in East Africa, Central Africa, and West Africa, while cold spots were observed across all Sub-Saharan African countries. The observed hotspots of open defecation in East and West African countries may be explained by a combination of factors.

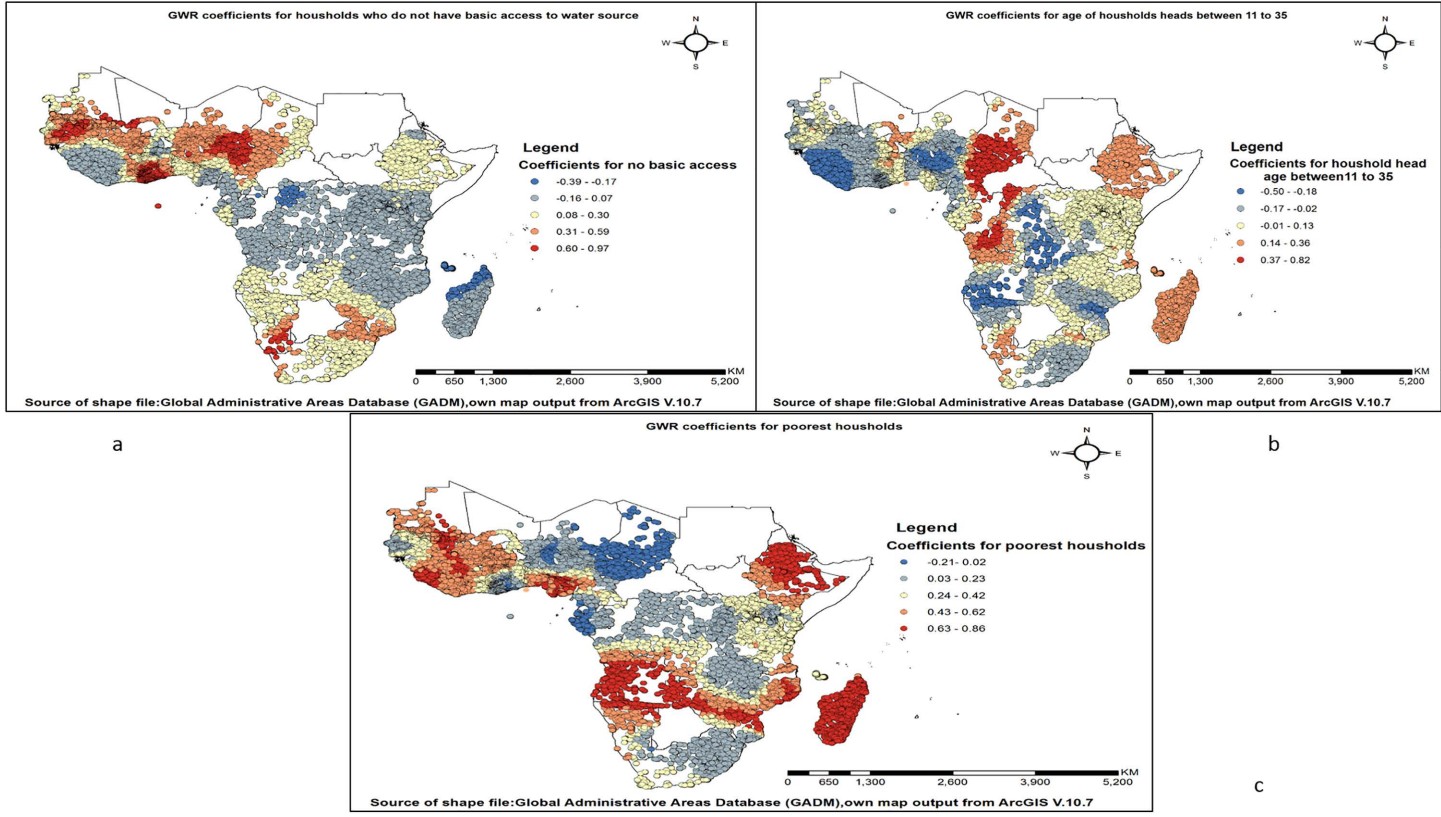

**Fig 7. Spatial variation in socioeconomic and demographic predictors associated with open defecation practice among households in sub-Saharan Africa.** (a) Coefficients for households with no basic access to water sources; (b) Coefficients for households headed by individuals aged 11–35 years; and (c) Coefficients for the poorest households.

These include entrenched cultural norms that accept or even prefer open defecation, low prioritization of sanitation infrastructure in rural development policies, limited access to affordable sanitation technologies, and widespread poverty [57]. In contrast, cold spots across Sub-Saharan Africa result from effective sanitation programs, urbanization, and socio-economic improvements [2,21,58,59].

Similarly, the Anselin Local Morans identified high and low clusters. High clusters were observed in Mozambique, Zambia, Zimbabwe, Chad, Madagascar, Ethiopia, Angola, Namibia, the Democratic Republic of the Congo, Niger, Nigeria, Mali, Côte d'Ivoire, Benin, Namibia, and Togo. Low clusters were also observed in South Africa, Ethiopia, Kenya, the Democratic Republic of the Congo, Rwanda, Uganda, Tanzania, Malawi, Zambia, Mozambique, Comoros, Cameroon, Gabon, Mali, Mauritania, Guinea, Senegal, Sierra Leone, Chad, Madagascar, Namibia, Angola, and Côte d'Ivoire. High OD clusters are driven by poverty, inadequate sanitation infrastructure, and cultural beliefs [56]. Many communities lack access to functional toilets, and cultural norms sometimes discourage public toilet use [21,60]. Conversely, low OD clusters result from effective sanitation programs, urbanization, and socio-economic improvements [58,61]. Urban centres provide better sanitation infrastructure, while improved economic conditions and education contribute to lower OD rates [21,61].

The SaTScan cluster analysis identified 146 windows containing significant clusters of households in Sub-Saharan Africa. Significant clusters were observed across West, East, Central, and Southern regions of sub-Saharan Africa. However, areas with the highest log likelihood values (ranging from 5659.39 to 9025.88) were found in Benin, Togo, and Ghana. The high log-likelihood values for open defecation (OD) observed in Benin, Togo, and Ghana can be attributed to

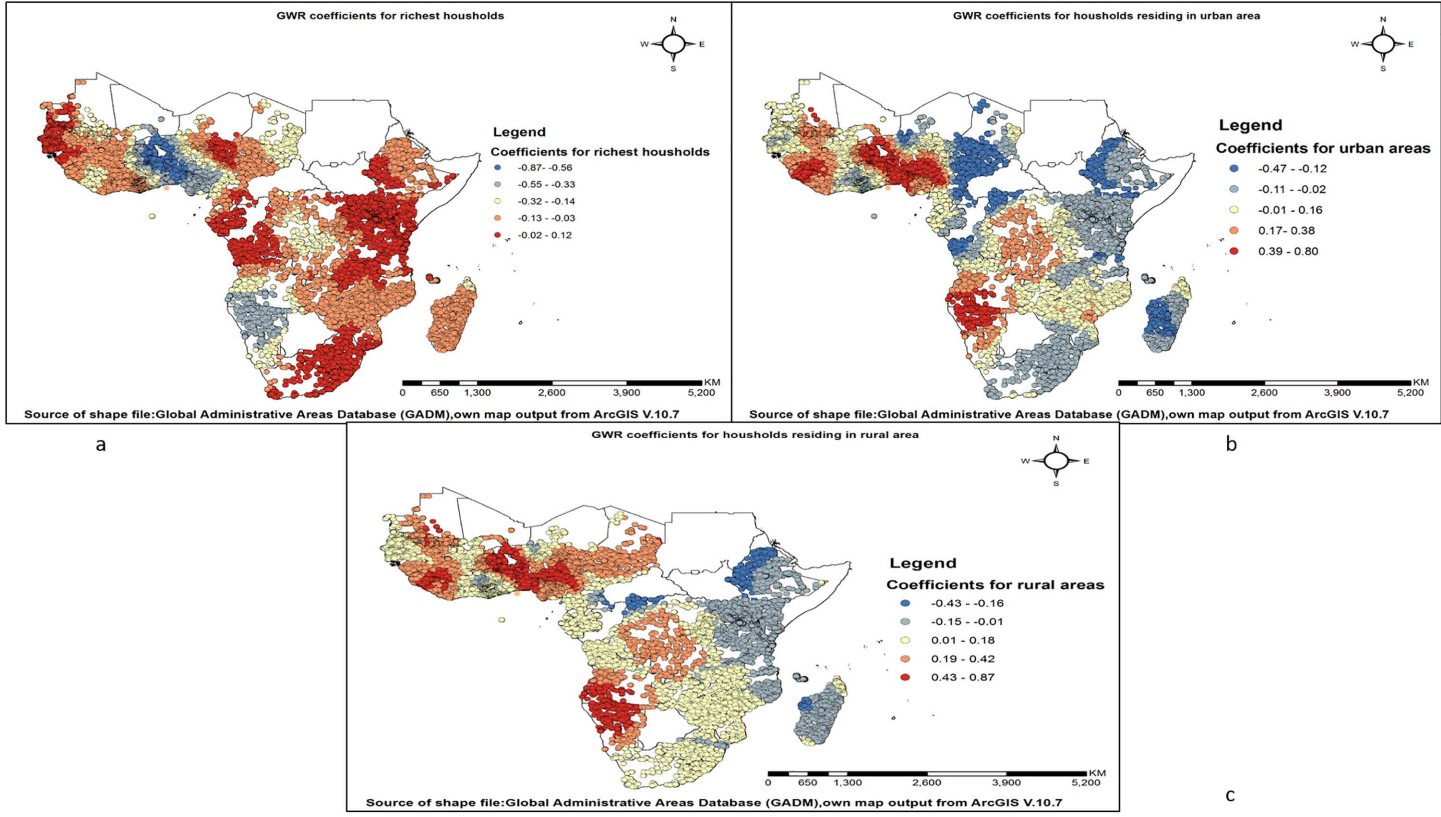

**Fig 8. Spatial variation in wealth and residence-related predictors associated with open defecation practice among households in sub-Saharan Africa.** (a) Coefficients for the richest households; (b) Coefficients for households in urban areas; and (c) Coefficients for households in rural areas.

several interrelated factors. Historically, these countries have experienced high prevalence rates of OD, with Benin (63%) and Togo (53%) having OD rates exceeding 50% in 2005, and Ghana also showing significant levels of OD [36,58,62]. A substantial portion of the population in these countries lacks access to basic sanitation facilities, with approximately 32% of rural households in Ghana practising OD due to limited access to proper sanitation [62]. Similarly, about 49% of the population in Benin engages in OD [36]. Additionally, cultural norms and beliefs play a significant role in influencing sanitation behaviours. In Ghana, inadequate access to sanitation, cultural beliefs, and investment priorities that favour drinking water over sanitation contribute to the persistence of OD practices [62].

The spatial regression analysis revealed that socio-demographic, economic, and water-related factors influence open defecation practices in Sub-Saharan Africa. Regression analysis showed that households with no formal education have a positive relationship with open defecation practice. Countries such as Chad, Nigeria, Ghana, Niger, the Democratic Republic of the Congo, Ethiopia, Kenya, Namibia, and Guinea have regions with the strongest positive correlation between open defecation practices and lack of education. A study conducted in Ghana and Ethiopia also highlighted this [54,63]. Households lacking formal education are more likely to practice open defecation due to limited awareness of proper sanitation and hygiene practices. A study published in Haiti found that lower education levels are significantly associated with higher rates of open defecation, highlighting the crucial role of education in promoting improved sanitation behaviours [59].

GWR analysis also revealed that households with more than four family members are negatively associated with open defecation practices. South Africa, Mozambique, the Democratic Republic of the Congo, Chad, Cameroon, Nigeria,

Burkina Faso, Ghana, Niger, Togo, Benin, and Ethiopia have areas with a negative association between open defecation and larger family size. The finding that households with more than four family members are negatively associated with open defecation practices, as shown by Geographically Weighted Regression (GWR) analysis, can be explained by several factors. Larger families often pool resources, making it easier to construct and maintain sanitation facilities [64]. Economies of scale also reduce the cost per person for sanitation investments, making it more feasible for larger households to invest in toilets [65].

Additionally, stronger social norms and peer pressure within larger families may encourage better sanitation practices [66]. Larger households may also have more educated members, increasing awareness of health risks related to open defecation [67]. Such kind of families are often targeted by government or NGO sanitation programs, which help reduce open defecation [68]. Moreover, in many sub-Saharan African societies, larger families tend to live in compound-style or extended households where sanitation facilities are shared among members, reducing the need for open defecation [69]. Furthermore, these households may have more diverse age groups, including older members or children, which can encourage more protective and normative sanitation behaviors [70]. Larger family size may be associated with higher household income due to the potential for multiple income earners, which in turn can enhance the household's creditworthiness. Improved credit access can facilitate the acquisition of targeted WASH loans or support investments in sanitation infrastructure. This economic advantage may also help explain the observed negative association between households with more than four members and open defecation practices.

Households who have unimproved drinking water sources and without basic access to a water source have a positive association with open defection practice in GWR analysis. The strongest positive correlation between open defecation practice and households with unimproved drinking water was observed in Comoros and Madagascar. The highest positive correlation between open defecation practice and households lacking basic access to a water source was observed in South Africa, Namibia, Nigeria, Cameroon, Ghana, Benin, Togo, Mali, Mauritania, Senegal, and Gambia. The positive association between households with unimproved drinking water sources and open defecation practices, as revealed by Geographically Weighted Regression (GWR) analysis, can be explained by several interrelated factors supported by existing research. Households with unimproved water sources and without basic access to a water source often lack access to sanitation facilities due to the interconnected nature of water, sanitation, and hygiene (WASH) interventions [71]. While larger family size was associated with lower open defecation prevalence in several regions via GWR analysis, limited water access appears to undermine this effect. Empirical evidence from multiple WASH studies indicates that households facing long distances to fetch water or relying on unimproved sources are more likely to practice open defecation. Limited water availability may constrain sanitation behaviors espite larger households' potential for resource pooling [72,73].

Households headed by individuals aged between 11 and 35 are more likely to practice open defecation. The highest positive correlation between open defecation practice and households headed by individuals in this age group was observed in Angola, the Democratic Republic of the Congo, Chad, and Cameroon. Younger household heads may have limited financial resources and less experience in managing household infrastructure, including sanitation facilities [74]. In addition, younger individuals may have less exposure to sanitation education and community health programs, contributing to poor sanitation practices which may lead to a high percentage of open defecation practices [75].

In Sub-Saharan Africa, the poorest households showed a strong positive association with open defecation, with the highest correlation observed in Ethiopia, Madagascar, Mozambique, Zimbabwe, Zambia, Angola, Namibia, Nigeria, Mauritania, Mali, Côte d'Ivoire, and Sierra Leone, where financial constraints and limited access to sanitation facilities contribute to the practice. The strong link between poverty and open defecation in Sub-Saharan Africa is driven by financial constraints, limited sanitation infrastructure, and low education levels. Poor households often lack the resources to build or maintain toilets, while many live in areas without proper sanitation facilities [21,76].

The richest households showed a negative association with open defecation, with the strongest negative association observed in Nigeria, Togo, Benin, Burkina Faso, and Ghana. The negative association between the richest households

and open defecation can be attributed to several factors. Wealthier households have the financial capacity to invest in improved sanitation facilities [76]. They also tend to live in areas with better infrastructure and sanitation services, as noted in the WHO/UNICEF (2017) Joint Monitoring Programme report [77]. Higher education levels among wealthier families contribute to greater awareness of the health risks of open defecation [78]. Additionally, wealthier households are more influenced by modern social and cultural norms that encourage hygienic practices [79].

Households residing in urban areas show a negative association with open defecation practices. The strongest negative association between open defecation and urban households were observed in Ethiopia, Madagascar, Chad, Tanzania, the Democratic Republic of the Congo, Cameroon, and Niger. Urban residents often have higher levels of education and income, which can lead to greater awareness of hygiene practices and a stronger ability to afford sanitation facilities [80]. Urban areas generally have better access to sanitation facilities such as toilets and latrines. The development of sanitation infrastructure in urban centres significantly reduces the prevalence of open defecation. According to a report by WHO/UNICEF (2017), urban areas tend to have higher coverage of sanitation facilities, which is linked to lower rates of open defecation [77,81].

Similarly, households residing in rural areas showed a negative association with open defecation practices, with the strongest negative correlations observed in Ethiopia, Madagascar, the Democratic Republic of the Congo, and Cameroon. Rural areas may benefit from community-led sanitation programs or government interventions that aim to promote hygienic practices [82]. This counterintuitive finding may reflect the impact of targeted rural sanitation programs, such as Community-Led Total Sanitation (CLTS), which have been implemented in many rural regions to promote behavioral change and community ownership of sanitation solutions. These localized interventions may lead to reduced open defecation rates even in rural settings, explaining the observed negative association in specific geographic contexts [83,84].

This study is based on cross-sectional data, which limits the ability to infer causal relationships between the identified predictors and open defecation practices. Additionally, there may be unmeasured confounding factors such as local sanitation campaigns, community enforcement of hygiene norms, seasonal variation, or cultural attitudes toward open defecation that were not included in the DHS dataset but may influence the outcomes.

## Conclusion and recommendations

This study is among the first to apply geographically weighted regression (GWR) to investigate spatial variations in household open defecation across sub-Saharan Africa. The study shows that the prevalence of open defecation remains high at 23.24%. Much work remains to be done to achieve Sustainable Development Goal (SDG) 6, specifically Target 6.2, which aims for an open defecation-free world by 2030. In addition, this study showed that open defecation practices varied across geographic regions in Sub-Saharan Africa, based on the Hotspot, Cluster and Outlier Analysis, and SaTScan cluster analysis. A key finding from the GWR analysis was that open defecation practices were significantly associated with households that had no formal education, larger family sizes (more than four members), use of unimproved water sources, lack of basic access to water, and household heads aged between 11 and 35 years. Furthermore, households in both the poorest and richest wealth quintiles, as well as those located in urban and rural areas, demonstrated significant spatial associations with open defecation.

To achieve target 6.2 of SDG 6, it will be necessary to promote targeted interventions to eliminate open defecation by 2030. Health-related education and awareness campaigns for sanitation should cater to uneducated households. Expansion of access to water and sanitation infrastructure must be ensured in urban as well as rural areas, giving priority to disadvantaged communities. Scale-up community-led sanitation programs, such as CLTS, which address the effects of poverty and large family sizes. Young household heads aged 11–35 must receive specific attention through youth-focused education and capacity-building activities. The governments must put in place policies to provide affordable sanitation facilities and encourage public-private partnerships. Upgrading informal settlements in cities and promoting low-cost, culturally appropriate sanitation facilities in rural areas is, therefore, of paramount importance.

Given the spatial clustering of open defecation in specific hotspots particularly in West and East Africa targeted policy interventions are essential. Governments and development partners should prioritize sanitation investments in high burden areas using geographically targeted approaches. Community-based programs such as CLTS, combined with infrastructure development and behavior change communication, should be scaled up in these regions. Special attention should also be given to reaching marginalized groups in under-served rural and peri-urban communities. Future research should investigate causal links between education, water access, and demographics using longitudinal and mixed methods. Finer-scale spatial analyses will improve intervention targeting and support achieving SDG 6.2 by 2030 for an open defecation-free sub-Saharan Africa.

## Strengths

This study employed advanced spatial analysis techniques help identify clusters of open defecation practice, highlighting areas in need of intervention. By considering both individual and community-level factors, the study provides a comprehensive understanding of open defection practice, enabling targeted policy recommendations

## Limitations

Some countries were excluded from the analysis due to the unavailability or outdated nature of DHS data. As a result, the coverage may be limited, potentially leading to the underrepresentation of open defecation practices in high-risk or underserved populations and limiting the generalizability of the findings to those contexts. Additionally, the DHS data used in this study were collected across different years between 2012 and 2024. These temporal variations may introduce inconsistencies when comparing countries, as sanitation conditions and interventions may have changed significantly over time. The cross-sectional nature of the data limits our ability to adjust for or account for these temporal differences.

## Supporting information

**S1 File. Supplementary country-level statistics.**
(DOCX)

## Author contributions

**Conceptualization:** Lidetu Demoze.

**Data curation:** Lidetu Demoze, Mitkie Tigabie, Gelila Yitageasu.

**Formal analysis:** Lidetu Demoze, Natnael Gizachew, Eshetu Abera Worede, Eyob Akalewold Alemu, Gelila Yitageasu.

**Investigation:** Lidetu Demoze, Gelila Yitageasu.

**Methodology:** Lidetu Demoze, Eshetu Abera Worede, Adem Tsegaw Zegeye, Abiy Ayele Angelo, Gelila Yitageasu.

**Project administration:** Lidetu Demoze, Gelila Yitageasu.

**Resources:** Lidetu Demoze.

**Software:** Lidetu Demoze, Natnael Gizachew, Adem Tsegaw Zegeye, Yayeh Walle Molalign, Mitkie Tigabie, Abiy Ayele Angelo, Eyob Akalewold Alemu, Gelila Yitageasu.

**Supervision:** Lidetu Demoze, Natnael Gizachew, Eshetu Abera Worede, Adem Tsegaw Zegeye, Yayeh Walle Molalign, Mitkie Tigabie, Eyob Akalewold Alemu, Gelila Yitageasu.

**Validation:** Lidetu Demoze, Abiy Ayele Angelo, Gelila Yitageasu.

**Visualization:** Lidetu Demoze, Gelila Yitageasu.

**Writing – original draft:** Lidetu Demoze.

**Writing – review & editing:** Lidetu Demoze, Gelila Yitageasu.

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
