## [Decision Letter · Decision Letter 0]

24 Jul 2025

Dear Dr. Demoze,

We look forward to receiving your revised manuscript.

Kind regards,

Dereje Oljira Donacho, PhD

Academic Editor

PLOS ONE

Journal Requirements:

2. Please note that your Data Availability Statement is currently missing [the repository name and/or the DOI/accession number of each dataset OR a direct link to access each database]. If your manuscript is accepted for publication, you will be asked to provide these details on a very short timeline. We therefore suggest that you provide this information now, though we will not hold up the peer review process if you are unable.

3. Natural Earth (public domain): http://www.naturalearthdata.com/ " xlink:type="simple"> We note that Figures 1,  3, 4, 5, 6, 7, 8, 9, 10, 11, 12, 13, 14, 15 and 16 in your submission contain map/satellite images which may be copyrighted. All PLOS content is published under the Creative Commons Attribution License (CC BY 4.0), which means that the manuscript, images, and Supporting Information files will be freely available online, and any third party is permitted to access, download, copy, distribute, and use these materials in any way, even commercially, with proper attribution. For these reasons, we cannot publish previously copyrighted maps or satellite images created using proprietary data, such as Google software (Google Maps, Street View, and Earth). For more information, see our copyright guidelines: http://journals.plos.org/plosone/s/licenses-and-copyright.

a. You may seek permission from the original copyright holder of Figures 1,  3, 4, 5, 6, 7, 8, 9, 10, 11, 12, 13, 14, 15 and 16 to publish the content specifically under the CC BY 4.0 license.

Reviewers' comments:

Reviewer's Responses to Questions

**Comments to the Author**

1. Is the manuscript technically sound, and do the data support the conclusions?

Reviewer #1: Yes

Reviewer #2: Yes

2. Has the statistical analysis been performed appropriately and rigorously?

Reviewer #1: Yes

Reviewer #2: Yes

3. Have the authors made all data underlying the findings in their manuscript fully available?

Reviewer #1: Yes

Reviewer #2: Yes

4. Is the manuscript presented in an intelligible fashion and written in standard English?

Reviewer #1: Yes

Reviewer #2: Yes

Reviewer #1: Comments to the Author

Technical Soundness & Data Support for Conclusions

The study is technically sound, with data largely supporting conclusions. However, clarify causality limitations due to cross-sectional design and address counterintuitive findings

Statistical Rigor

Assessment:-

Generally acceptable with minor clarifications needed

Spatial analyses are robust, but provide:-

Timeframe of DHS data per country

Multicollinearity assessment in GWR

Criteria for cluster/household inclusion.

Methodological Standards

Adheres to spatial epidemiology standards but requires:-

Enhanced documentation of variable selection for GWR

Justification for bandwidth selection in spatial analyses.

Conclusion Support

Conclusions align with results, but strengthen by:-

Discussing drivers behind spatial clusters

Integrating policy implications for identified hotspots.

Data Repository Compliance

Data availability statement is appropriate, citing DHS program access. Ensure supplementary country-level statistics are included.

Presentation & Language

Generally intelligible but requires:-

Minor English editing for conciseness

Higher-resolution maps with clear legends/scales.

Detailed Feedback

Strengths

Addresses critical public health challenge aligned with SDG 6

Rigorous spatial methodology with large representative sample

Clear structure and contextualization.

Major Issues

Methods Detail:-

Specify temporal range of DHS datasets used.

Detail handling of missing data and variable coding.

Please clarify the time frame of the DHS data included for each country. Specify inclusion/exclusion criteria for clusters and households, and provide more detail on how variables were selected and handled in the GWR, including how you addressed multicollinearity

Interpretation of Spatial Results and Conclusion:-

Assessment:-Valid but requires more cautious language

The conclusion that larger households are negatively associated with open defecation may seem counterintuitive and could benefit from referencing cultural/family dynamics or sanitation sharing behavior.

Statements like “richest households negatively associated” need caution:-correlation ≠ causation, and the pathways should be clarified.

Explain mechanisms driving East/West African hotspots beyond statistical identification.

The spatial analyses are robust, but the discussion would benefit from deeper interpretation of the drivers behind identified hotspots and coldspots, and more explicit policy implications for regions with high prevalence

Causal Inference and Confounding:-

Emphasize cross-sectional limitations regarding causal inference in discussion.

Discuss potential unmeasured confounders

Urban vs. Rural Residency:-

The finding that both urban and rural residency are negatively associated with open defecation is counterintuitive. Please clarify this result and discuss possible explanations or limitations in the data or analytic approach.

Minor Issues

Tables/Figures:-

Ensure that all figures/maps are high-resolution and include legends and scales for clarity.

Ethics and Data Statements:-

Assessment:-Fully compliant

The ethics and data availability statements are appropriate and transparent, but consider explicitly listing the countries and IRBs where possible IRBs.

Clearly stated that this is a secondary analysis of DHS data, which had ethical clearance and informed consent in all countries.

Language:-

The manuscript is generally clear but would benefit from minor English language editing for conciseness and flow.

Suggestions

Include a limitations subsection addressing:-

Regional data gaps (e.g., unstable regions not covered)

Temporal variations in DHS collection.

Expand policy recommendations targeting hotspot interventions.

Comments to the Authors

Strengths

The study addresses a major public health and development challenge in sub-Saharan Africa, with direct relevance to SDG 6.

Use of large representative DHS datasets and advanced spatial methods adds rigor and regional nuance.

The Study is generally well-structured, with clear aims, methods, and results.

Reviewer #2: Reviewers comment

The article investigates “The Prevalence, Spatial Distribution, and Geographic Weighted Regression of Open Defecation Practices in sub-Saharan Africa Using Demographic and Health Survey (DHS) Data.” The authors establish a strong foundation for their study by clearly identifying research gaps and employing appropriate methodologies to address these deficiencies. The sample size is adequate, and the data analysis tools used are sufficient. The methodology of the study is robust, ensuring reproducibility. Overall, it offers valuable insights into open defecation in sub-Saharan Africa, providing evidence that could inform WASH interventions, policy, and practice. However, the authors do not emphasise the key novel contributions of the study. Although a wealth of results is presented, the discussions fail to thoroughly build on the results, explore the practical implications of these findings and how they can be applied to stimulate interventions to effectively reduce open defecation. I suggest authors improve the level of English used for the write-up.

Specific comments:

1.The section on abstract: The abstract gives a summary of the study, providing essential details such as the basis for the study, sample size, methodology, results, and discussions. The abstract can be strengthened by the following modifications.

•The conclusion is a repeat of the methodology and the last sentence of the background. The conclusion should highlight the key finding(s) and their implication on policy and future research. What makes this study novel?

•The study asserts 23.24% of open defecation practices in Sub-Saharan Africa. How does this compare to the value obtained by the JMP? This should be highlighted briefly.

2.The section on introduction: The introduction provides a good background to the study, citing relevant literature and establishing the basis for the study. However, the write-up is disjointed and lacks chronology. Literature on prevalence, access to WASH services, and global and regional contexts of OD has been intertwined. It is recommended that the introduction be rewritten to follow a chronology of thought. Minor comments include:

•There should be spacing between the square brackets and words. Eg. …global crises [1] NOT global crises[1]. This seems to appear throughout the write-up.

•Line 71-71 It would be essential to state the most current prevalence of OD in SSA. What is the prevalence?

•Line 77-78 Authors state OD is prevalent in parts of SSA? Authors should state or give examples of which countries of SSA.

3.The section on study area:

•Line 118 to 123 should be added as part of the introduction

4.The section on methodology: The methodology discussed is robust and reproducible. This gives credence to the data collected.

5.The section on Results and Discussion:

•Line 373 Separate bracket from “Leone”

•Lines 406 to 409: Authors provide reasons for the disparity in OD observed in SSA, Haiti and Indonesia. The reasons given are generic but written as though empirical. Authors should reword based on available evidence.

•Line 420 Check punctuations. Separate punctuations from words.

•Lines 430 to 436, Ghana is missing from the hotspot and coldspot classification. The authors measured access to water, and it will be interesting to highlight how this corresponds to the coldspot and hotspot classification of the OD region.

•Line 443-44 “Significant clusters were observed across nearly all regions of the region” should be reworded for clarity.

•Line 467 The GWR analysis predicted the influence of family size on OD. It would be interesting to triangulate family size, access to water and OD.

•Lines 467 to 480 Family size has a bearing on family income. Family income level determines creditworthiness and, by extension, access to targeted WASH loans for sanitation infrastructure. This could be a reason for the disparity in OD observed for different family sizes within the regions. Authors can consider this in the discussion.

oAge bracket of head of household is too wide (11 to 35 years). Better inferences can be drawn if the age brackets are reduced further into more categories. For example, lower age brackets can influence inferences on household size, income, access to water and OD.

6.The section on Conclusion:

•The conclusions and recommendations seem to be an extension of the results and discussion. Authors must only highlight the novelty of the study, the key findings, and their implications for policy, practise, and future research.

7.Figures

•For Fig 1: Countries like Seychelles, Equatorial Guinea, Mauritius and Cape Verde have been duplicated

**Do you want your identity to be public for this peer review?** For information about this choice, including consent withdrawal, please see our Privacy Policy

Reviewer #1: **Yes: ** Achiso, Yisihak

Reviewer #2: No

---

## [Author Response · Author response to Decision Letter 1]

1 Aug 2025

Responses to the Editors and review’s comments

Dear PLOS ONE editorial team,

Thank you for giving us the opportunity to submit a revised draft of the manuscript, and we would also like to thank you for your crucial comments on our paper (Manuscript ID: PONE-D-25-12133). We are very concerned and have combined all the suggested comments provided, which we believe strengthen our paper, and we hope this will render our paper eligible for consideration for publication in your reputed journal. We appreciate the time and effort that you and the reviewers dedicated to providing feedback on our manuscript and are grateful for the insightful comments and valuable improvements to our paper for publication.

The authors would like to inform you that we have addressed the comments and recommendations of the handling editor point by point. In addition, throughout our revision, we made our best corrections too. All changes made to the original version are highlighted using tracking changes and attached as “Revised Manuscript with Track Changes”. The unmarked copy of the manuscript is also attached as “Manuscript”. In addition, please see below a rebuttal letter that responds to each point raised by the handling editor, and this letter is also attached to the submission as “Response to Reviewers”.

Response to editor’s comments

Comments from the handling editor:

Author’s response: Dear Editor, thank you very much for your recommendation. We have made the corrections accordingly to meet the journal requirements.

2. Please note that your Data Availability Statement is currently missing [the repository name and/or the DOI/accession number of each dataset OR a direct link to access each database]. If your manuscript is accepted for publication, you will be asked to provide these details on a very short timeline. We therefore suggest that you provide this information now, though we will not hold up the peer review process if you are unable. In your revision ensure you cite all your sources (including your own works), and quote or rephrase any duplicated text outside the methods section. Further consideration is dependent on these concerns being addressed.

Author’s response: Dear Editor, thank you very much for your recommendations. We have made the necessary corrections to meet the journal's requirements.

3. We note that Figures 1, 3, 4, 5, 6, 7, 8, 9, 10, 11, 12, 13, 14, 15 and 16 in your submission contain map/satellite images which may be copyrighted. All PLOS content is published under the Creative Commons Attribution License (CC BY 4.0), which means that the manuscript, images, and Supporting Information files will be freely available online, and any third party is permitted to access, download, copy, distribute, and use these materials in any way, even commercially, with proper attribution. For these reasons, we cannot publish previously copyrighted maps or satellite images created using proprietary data, such as Google software (Google Maps, Street View, and Earth). For more information, see our copyright guidelines: http://journals.plos.org/plosone/s/licenses-and-copyright.

a. You may seek permission from the original copyright holder of Figures 1, 3, 4, 5, 6, 7, 8, 9, 10, 11, 12, 13, 14, 15 and 16 to publish the content specifically under the CC BY 4.0 license.

Author’s response: Dear Editor, thank you very much for your recommendation. We have made the corrections accordingly to meet the journal requirements.

This is Global Administrative Areas Database (GADM) license

The data are freely available for academic use and other non-commercial use. Redistribution or commercial use is not allowed without prior permission. Using the data to create maps for publishing of academic research articles is allowed. Thus you can use the maps you made with GADM data for figures in articles published by PLoS, Springer Nature, Elsevier, MDPI, etc. You are allowed (but not required) to publish these articles (and the maps they contain) under an open license such as CC-BY as is the case with PLoS journals and may be the case with other open access articles. Data for the following countries is covered by a a different license Austria: Creative Commons Attribution-Share Alike 2.0 (source: Government of Ausria)

You can find this information through this link: https://gadm.org/license.html

In addition previous study also used maps from GADM here it is the link: https://journals.plos.org/plosone/article?id=10.1371/journal.pone.0318189

Author’s response: Dear Editor, thank you very much for your recommendation. We have made the corrections accordingly to meet the journal requirements.

Comments from Reviewer #1:

1. The study is technically sound, with data largely supporting conclusions. However, clarify causality limitations due to cross-sectional design and address counterintuitive findings

Author’s response: Thank you very much for your comment and suggestions. We have made corrections accordingly.

2. Generally acceptable with minor clarifications needed. Spatial analyses are robust, but provide:- Timeframe of DHS data per country

Author’s response: Thank you very much for your comment and suggestions. We have made corrections accordingly.

3. Multicollinearity assessment in GWR, Criteria for cluster/household inclusion.

Author’s response: Thank you very much for your comment and suggestions. We have made corrections accordingly. We have included statement in the “Spatial regression analysis” section of the manuscript.

Clusters and households were included in the analysis if they had complete data on the outcome variable (open defecation status) and all selected explanatory variables. The analysis was conducted using the Household Recode (HR) files from the DHS datasets. Clusters with missing spatial coordinates or implausible values were excluded to maintain spatial accuracy.

The sanitation variable was extracted from the HR file using the DHS variable code hv205, which indicates the type of toilet facility used by the household. This variable was recoded into a binary outcome: households were classified as practicing open defecation (Yes) if hv205 = no facility/bush/field, and No otherwise (i.e., any other improved or unimproved facility

4. Data Repository Compliance. Data availability statement is appropriate, citing DHS program access. Ensure supplementary country-level statistics are included.

Author’s response: Thank you very much for your comment and suggestions. We have made corrections accordingly. We have included country-level statistics as S1 File.

5. Presentation & Language Generally intelligible but requires: - Minor English editing for conciseness Higher-resolution maps with clear legends/scales.

Author’s response: Thank you very much for your comment and suggestions. We have made corrections accordingly.

6. Specify temporal range of DHS datasets used.

Author’s response: Thank you very much for your comment and suggestions. We have made corrections accordingly

7. Detail handling of missing data and variable coding.

Author’s response: Thank you very much for your comment and suggestions. We have made corrections accordingly in data monument and analysis section of the manuscript.

8. Please clarify the time frame of the DHS data included for each country. Specify inclusion/exclusion criteria for clusters and households, and provide more detail on how variables were selected and handled in the GWR, including how you addressed multicollinearity

Author’s response: Thank you very much for your comment and suggestions. We have made corrections accordingly

9. The conclusion that larger households are negatively associated with open defecation may seem counterintuitive and could benefit from referencing cultural/family dynamics or sanitation sharing behavior.

Author’s response: Thank you very much for your comment and suggestions. We have made corrections accordingly

10. Statements like “richest households negatively associated” need caution:-correlation ≠ causation, and the pathways should be clarified.

Author’s response: Thank you very much for your comment and suggestions. We have made corrections accordingly. We change the phrase from correlation to negative association.

11. Explain mechanisms driving East/West African hotspots beyond statistical identification.

Author’s response: Thank you very much for your comment and suggestions. We have made corrections accordingly.

12. The spatial analyses are robust, but the discussion would benefit from deeper interpretation of the drivers behind identified hotspots and coldspots, and more explicit policy implications for regions with high prevalence

Author’s response: Thank you very much for your comment and suggestions. We have made corrections accordingly is discussion section of manuscript.

13. Emphasize cross-sectional limitations regarding causal inference in discussion. Discuss potential unmeasured confounders

Author’s response: Thank you very much for your comment and suggestions. We have made corrections accordingly is discussion section of manuscript..

14. The finding that both urban and rural residency are negatively associated with open defecation is counterintuitive. Please clarify this result and discuss possible explanations or limitations in the data or analytic approach.

Author’s response: Thank you very much for your comment and suggestions. We have made corrections accordingly.

15. Tables/Figures:- Ensure that all figures/maps are high-resolution and include legends and scales for clarity.

Author’s response: Thank you very much for your comment and suggestions. We have made corrections accordingly and all figures/maps complies with journal guideline.

16. Ethics and Data Statements:- Assessment:-Fully compliant. The ethics and data availability statements are appropriate and transparent, but consider explicitly listing the countries and IRBs where possible IRBs. Clearly stated that this is a secondary analysis of DHS data, which had ethical clearance and informed consent in all countries.

Author’s response: Thank you very much for your comment and suggestions. We have made corrections accordingly.

17. Language:- The manuscript is generally clear but would benefit from minor English language editing for conciseness and flow.

Author’s response: Thank you very much for your comment and suggestions. We have made some corrections accordingly.

18. Suggestions Include a limitations subsection addressing:- Regional data gaps (e.g., unstable regions not covered) Temporal variations in DHS collection. Expand policy recommendations targeting hotspot interventions.

Author’s response: Thank you very much for your comment and suggestions. We have made some corrections accordingly. We have included a statement in the recommendation and limitation section of the manuscript.

Comments from Reviewer #2:

1. The conclusion is a repeat of the methodology and the last sentence of the background. The conclusion should highlight the key finding(s) and their implication on policy and future research. What makes this study novel?

Author’s response: Thank you very much for your comment and suggestions. We have made some corrections accordingly. We included a statement In conclusion section of the manuscript.

2. The study asserts 23.24% of open defecation practices in Sub-Saharan Africa. How does this compare to the value obtained by the JMP? This should be highlighted briefly.

Author’s response: Thank you very much for your comment and suggestions. We have made some corrections accordingly. We included a statement In conclusion section of the manuscript.

3. There should be spacing between the square brackets and words. Eg. …global crises [1] NOT global crises[1]. This seems to appear throughout the write-up.

Author’s response: Thank you very much for your comment and suggestions. We have made some corrections accordingly.

4. Line 71-71 It would be essential to state the most current prevalence of OD in SSA. What is the prevalence?

Author’s response: Thank you very much for your comment and suggestions. We have made corrections accordingly in line number 95-96.

5. Line 77-78 Authors state OD is prevalent in parts of SSA? Authors should state or give examples of which countries of SSA.

Author’s response: Thank you very much for your comment and suggestions. We have made corrections accordingly in line number 78-81

6. The section on study area: Line 118 to 123 should be added as part of the introduction

Author’s response: Thank you very much for your comment and suggestions. We have made corrections accordingly.

7. Line 373 Separate bracket from “Leone”

Author’s response: Thank you very much for your comment and suggestions. We have made corrections accordingly.

8. Lines 406 to 409: Authors provide reasons for the disparity in OD observed in SSA, Haiti and Indonesia. The reasons given are generic but written as though empirical. Authors should reword based on available evidence.

Author’s response: Thank you very much for your comment and suggestions. We have made corrections accordingly.

9. Line 420 Check

---

## [Decision Letter · Decision Letter 1]

7 Oct 2025

Dear Dr. Demoze,

Thank you for submitting your manuscript to PLOS ONE. After careful consideration, we feel that it has merit but does not fully meet PLOS ONE’s publication criteria as it currently stands. Therefore, we invite you to submit a revised version of the manuscript that addresses the points raised during the review process.

**ACADEMIC EDITOR: **

2. Also, table 5 is not really necessarily in one specific table. You can put it under table 4.

We look forward to receiving your revised manuscript.

Kind regards,

D. Daniel, Ph.D.

Academic Editor

PLOS ONE

Journal Requirements:

Reviewers' comments:

Reviewer's Responses to Questions

**Comments to the Author**

Reviewer #2: All comments have been addressed

2. Is the manuscript technically sound, and do the data support the conclusions?

Reviewer #2: Yes

3. Has the statistical analysis been performed appropriately and rigorously?

Reviewer #2: Yes

4. Have the authors made all data underlying the findings in their manuscript fully available?

Reviewer #2: Yes

5. Is the manuscript presented in an intelligible fashion and written in standard English?

Reviewer #2: Yes

Reviewer #2: Line 522

increasing awareness of health risks related to open defecation [65].Such kind of families are...

Authors should space the full stop from the word "such"

**Do you want your identity to be public for this peer review?** For information about this choice, including consent withdrawal, please see our Privacy Policy

Reviewer #2: No

---

## [Author Response · Author response to Decision Letter 2]

18 Oct 2025

Responses to the Editors and review’s comments

Dear PLOS ONE editorial team,

Thank you for giving us the opportunity to submit a revised draft of the manuscript, and we would also like to thank you for your crucial comments on our paper (Manuscript ID: PONE-D- 25-12133R1). We are very concerned and have combined all the suggested comments provided, which we believe strengthen our paper, and we hope this will render our paper eligible for consideration for publication in your reputed journal. We appreciate the time and effort that you and the reviewers dedicated to providing feedback on our manuscript and are grateful for the insightful comments and valuable improvements to our paper for publication.

The authors would like to inform you that we have addressed the comments and recommendations of the handling editor point by point. In addition, throughout our revision, we made our best corrections too. All changes made to the original version are highlighted using tracking changes and attached as “Revised Manuscript with Track Changes”. The unmarked copy of the manuscript is also attached as “Manuscript”. In addition, please see below a rebuttal letter that responds to each point raised by the handling editor, and this letter is also attached to the submission as “Response to Reviewers”.

Response to editor’s comments

Comments from the handling editor:

1. I notice there are many figures, followed by 1-2 sentences explaining the figure, i.e., figure 5-16. Could you please combine 3-4 of them into two-three big figures, e.g., figure 5-8 to figure 5a-d (so 4 maps in 1 figure). And combine the paragraphs explaining those figures, so one paragraph does not consist only on 2 sentences.

Author’s response: Dear Editor, thank you very much for your recommendation. We have made the corrections accordingly to meet the journal requirements.

2. Also, table 5 is not really necessarily in one specific table. You can put it under table 4.

Author’s response: Dear Editor, thank you very much for your recommendation. We have made the corrections accordingly to meet the journal requirements.

Comments from Reviewer #1:

1. Line 522 ...increasing awareness of health risks related to open defecation [65].Such kind of families are... Authors should space the full stop from the word "such"

Author’s response: Thank you very much for your comment and suggestions. We have made corrections accordingly.

---

## [Editor Report · Decision Letter 2]

3 Nov 2025

The Prevalence, Spatial Distribution and Geographic Weighted Regression of Open Defecation Practice in sub-Saharan Africa Using Demographic and Health Survey (DHS) Data

PONE-D-25-12133R2

Dear Dr. Demoze,

We’re pleased to inform you that your manuscript has been judged scientifically suitable for publication and will be formally accepted for publication once it meets all outstanding technical requirements.

Kind regards,

D. Daniel, Ph.D.

Academic Editor

PLOS ONE
---

## [Editor Report · Acceptance letter]

PONE-D-25-12133R2

PLOS ONE

Dear Dr. Demoze,

I'm pleased to inform you that your manuscript has been deemed suitable for publication in PLOS ONE. Congratulations! Your manuscript is now being handed over to our production team.

Kind regards,

on behalf of

Dr. D. Daniel

Academic Editor

PLOS ONE